# REAL-TIME NEURAL VOICE CAMOUFLAGE

**Mia Chiquier, Chengzhi Mao, Carl Vondrick**
Department of Computer Science, Columbia University
New York, NY, 10025
{mac2500, cm3797, cv2428}@columbia.edu
voicecamo.cs.columbia.edu

## ABSTRACT

Automatic speech recognition systems have created exciting possibilities for applications, however they also enable opportunities for systematic eavesdropping. We propose a method to camouflage a person's voice over-the-air from these systems without inconveniencing the conversation between people in the room. Standard adversarial attacks are not effective in real-time streaming situations because the characteristics of the signal will have changed by the time the attack is executed. We introduce predictive attacks, which achieve real-time performance by forecasting the attack that will be the most effective in the future. Under real-time constraints, our method jams the established speech recognition system Deep-Speech 3.9x more than baselines as measured through word error rate, and 6.6x more as measured through character error rate. We furthermore demonstrate our approach is practically effective in realistic environments over physical distances.

## 1 INTRODUCTION

Automatic speech recognition models are embedded in nearly all smart devices. Although these models have many exciting applications, the concern for the potential of these devices to eavesdrop is significant. It is becoming increasingly important to develop methods that give users the autonomy to safeguard their speech from voice processing software.

Fortunately, over the last decade, there has been work demonstrating that neural networks models are easily fooled. For example, they remain vulnerable to small additive perturbations (Carlini & Wagner, 2018), ambient noise (Xu et al., 2020), and unusual examples (Nguyen et al., 2015). Predominant methods such as gradient-based methods and their variants have remained the standard approach to generating challenging examples for deep neural networks (Madry et al., 2019). However, to achieve this, these methods require the full input upfront, and thus users can not practically use them as they continuously speak.

Therefore, the community has increasingly been focusing on researching general, robust methods of breaking neural networks that can be used in real-time. We define *robust* to mean an obstruction that can not be easily removed, *real-time* to mean an obstruction that is generated continuously as speech

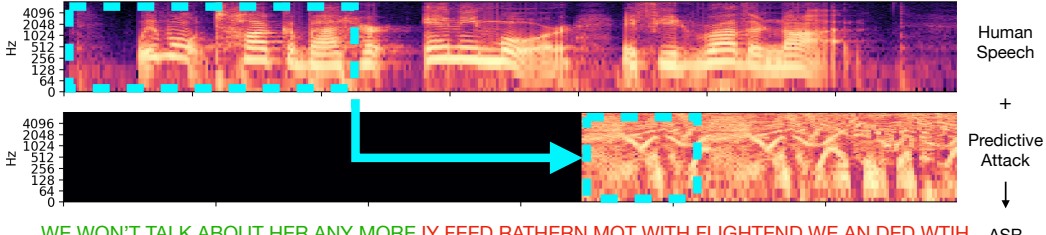

WE WON'T TALK ABOUT HER ANY MORE IY FEED RATHERN MOT WITH FLIGHTEND WE AN DED WTIH

Figure 1: We introduce "Neural Voice Camouflage," an approach that disrupts automatic speech recognition systems in real time. To operate on live speech, our approach must predict corruptions into the future so that they may be played in real-time. The method works for the majority of the English language. Green/red indicates correct/incorrect transcription respectively.

is spoken, and *general* to mean applicable to the majority of vocabulary in a language. Existing prior work has successfully tackled at least one of these three requirements, but none all three. While some work is real-time (Chen et al., 2020; Schönherr et al., 2018), these disruptions can be filtered out as they are constrained to specific frequency ranges. Universal attacks (Lu et al., 2021) can be similarly subtracted. Gong et al. (2019) achieved both real-time and robust obstructions, but are limited to a predefined set of ten words.

Streaming audio is a particularly demanding domain to disrupt because the calculation needs to be performed in real-time. By the time a sound is computed, time will have passed and the streaming signal will have changed, making standard generative methods obsolete. The sampling rate of audio is at least 16 kHz, meaning the corruption for a given input must be estimated and played over a speaker within milliseconds, which is currently infeasible. Additionally, when attacks are played over-the-air, the attack needs to be loud enough to disrupt any rogue microphone that could be far away. The attack sound needs to carry the same distance as the voice.

We introduce predictive attacks, which are able to disrupt any word that automatic speech recognition models are trained to transcribe. Our approach achieves real-time performance by *forecasting* an attack on the future of the signal, conditioned on two seconds of input speech. Our attack is optimized to have a volume similar to normal background noise, allowing people in a room to converse naturally and without monitoring from an automatic speech recognition system.

Forecasting with deep neural networks has already been successfully used in other domains to achieve real-time performance, for instance in packet loss concealment (Pascual et al., 2021). In this paper, we demonstrate how and why this approach lends itself particularly well to developing general, robust and real-time attacks for automatic speech recognition models. Our experiments show that predictive attacks are able to largely disrupt the established DeepSpeech (Amodei et al., 2016) recognition system which was trained on the LibriSpeech dataset (Panayotov et al., 2015). On the standard, large-scale dataset LibriSpeech, our approach causes at least a three fold increase in word error rate over baselines, and at least a six fold increase in character error rate.

Our method is practical and straightforward to implement on commodity hardware. We additionally demonstrate the method works inside real-world rooms with natural ambient noise and complex scene geometries. We call our method Neural Voice Camouflage.

## 2 RELATED WORK

**Breaking Neural Networks:** Szegedy et al. (2014) first discovered adversarial attacks in computer vision. Since then, a large number of methods to break neural networks have been introduced (Madry et al., 2019; Kurakin et al., 2017; Carlini & Wagner, 2017; Croce & Hein, 2020; Moosavi-Dezfooli et al., 2016; Goodfellow et al., 2014) , where noise optimized by gradient descent fool the state-of-the-art models. Audio adversarial attacks (Carlini & Wagner, 2018; Qin et al., 2019; Yakura & Sakuma, 2019; Schönherr et al., 2018) have also been constructed. Gradient based iterative adversarial attacks, while effective, are computationally intensive, and need to see the whole example first before launching the attack. Faster adversarial attacks uses generators to generate attacks (Xiao et al., 2019). However, the attacks are still offline. To make the adversarial attack reliable for live speech, the attacker needs to anticipate the future in an online manner.

**Online Attacks:** Real-time attacks are an emerging area of research in machine learning and there have been several initial works. For example, Gong et al. (2019) develop a reinforcement learning based approach to balance the trade-off between number of samples seen and attack deployment duration. They also optimize a volume trade-off to achieve over-the-air performance. While they learn to disrupt spoken keyword detection (a predefined set of ten words), our approach is able to obfuscate entire sentences. Further, attacks for streaming data with bayesian approaches have been proposed (Braverman et al., 2021; Seraphim & Poovammal, 2021). However, they are unable to tackle high-dimensional data such as audio. Another direction prior work has taken to create online attacks is to constantly be attacking a certain word (Li et al., 2019). Although this works in real-time, it only targets the wake word, and not full sentences. There also have been a few methods that jam microphones by emitting sound outside the range of human hearing. For example, Chen et al. (2020) developed an approach to emit ultrasonic attacks and Schönherr et al. (2018) also generate

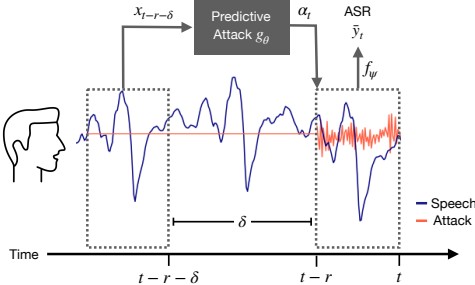

Figure 2: We illustrate our problem set-up for predictive attacks. In order to attack the audio starting at time $t - r$, we need to start computing the attack by time $t - r - \delta$, assuming it takes an upper bound of $\delta$ time to record, compute and play the attack. Our approach is able to obtain real-time performance by predicting this attack in the future, given the previous observations of the stream.

attacks outside the human hearing range. However, by limiting the attack to specific frequencies, a defender can design a microphone that filters this set of frequencies out.

**Robustness:** Due to the importance of this problem, there has been extensive research into learning robust models (Madry et al., 2019; Carmon et al., 2019; Wang et al., 2020; Mao et al., 2019; 2020; 2021). However, building defenses is challenging, and work has even shown that many basic defenses, such as adding randomness to the input, are not effective (Athalye et al., 2018). Among all the defense strategies, adversarial training proposed by Madry et al. (2019) is the standard defense that has been most widely used. However, adversarial training has the drawback that it improves robustness accuracy at the cost of reducing the original accuracy (Tsipras et al., 2019), which is the reason that adversarial training is not used in most real-world applications. In this paper, we show our approach is still effective against these established defenses.

**Real-time Machine Learning:** Interest in real-time artifical intelligence dates back to 1996, starting with anytime algorithms, which return a solution at any given point in time (Zilberstein, 1996). More recently, there have been challenges to evaluate vision models in real-time (Kristan et al., 2017). Generally, there has been a focus on speeding up forward passes to allow for faster inference, thereby approaching real-time (Howard et al., 2017). In addition, there has been recent work in leveraging deep neural network predictions to achieve real-time performance, which has been applied to speech packet loss concealment (Pascual et al., 2021). This differs from the previous approaches of improving inference speed. Recently, the community has recently taken an interest in establishing robust metrics and evaluations for real-time inference (Li et al., 2020).

## 3 METHOD

We present our approach for creating real-time obstructions to automatic speech recognition (ASR) systems. We first motivate the background for real-time attacks, then introduce our approach that achieves online performance through predictive attack models.

### 3.1 STREAMING SPEECH RECOGNITION

Let $x_t$ be a streaming signal that represents the input speech up until time $t$. The goal of ASR is to transcribe this signal into the corresponding text $y_t$. The field often estimates this mapping through a neural network $\hat{y}_t = f_\psi(x_t)$ where the parameters $\psi$ are optimized to minimize the empirical risk $\min_\psi \mathbb{E}_{(x,y)} \left[ \mathcal{L}\left(\hat{y}_t, y_t\right)\right]$. For modeling sequences, the CTC loss function is a common choice for $\mathcal{L}$ (Graves et al., 2006).

In offline setups, we can corrupt the neural network $f_\psi$ with a standard adversarial attack. These attacks work by finding a minimal additive perturbation vector $\alpha_t$ that, when added to the input signal, produces a high loss: $\arg\max_{\alpha_t} \mathcal{L}\left(f_\psi\left(x_t + \alpha_t\right), y_t\right)$ subject to a bound on the norm of the perturbation $\|\alpha_t\|_\infty < \epsilon$. Adversarial attacks, such as projected gradient descent (PGD) or fast gradient descent, have been widely effective on vision and speech datasets (Madry et al., 2019; Goodfellow et al., 2014; Carlini & Wagner, 2018). Defending against them both empirically and theoretically remains an active area of research today.

Standard adversarial attacks will optimize the perturbation vector $\alpha_t$ conditioned on the current position of the stream $x_t$. However, by the time the solution $\alpha_t$ is found for a stream, the attack will be obsolete because time will have passed and the condition will have almost certainly changed.

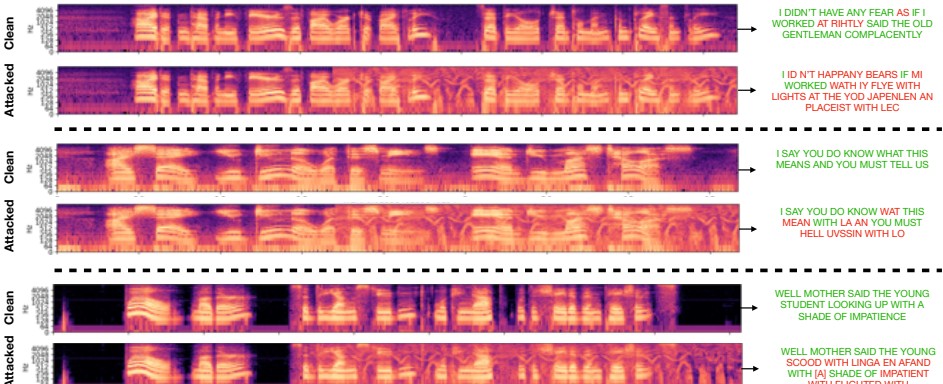

Figure 3: We illustrate three examples of our attack in action. The first row in a pair is the clean spectrogram of the input speech signal. The second row in the pair is the attacked spectrogram of the speech signal. We note that the difference becomes visible after 2.5s of the speech signal, since our method requires 2s of input and has a delay of ($\delta$) 0.5s before it can predict an attack. As seen, the predicted attack resembles that of speech formants.

Audio is a particularly demanding domain because the high sampling rate (as high as $48\,\mathrm{kHz}$) would require attacks to be computed nearly instantaneously (less than $20$ microseconds). Furthermore, applying the stale $\alpha_t$ to the future $x_{t+\delta}$ will not work because the attack vectors are optimized to corrupt the features of their input, which may vary over time.

## 3.2 PREDICTIVE REAL-TIME ATTACKS

We propose a class of predictive attacks, which enable real-time performance by forecasting the attack vector that will be effective in future time steps. It will invariably take some time for the attack to be computed. For attacks to operate in real-time environments, this means the attack needs to be optimized not for the observed signal, but for the unobserved signal in the future. If our observation of the signal $x_t$ is captured at time $t$ and our algorithm takes $\delta$ seconds to compute an attack and play it, then we need to attack the signal starting at $x_{t+\delta}$. However, creating these attacks is challenging practically because real-world signals are stochastic and multi-modal. Due to the high uncertainty, generating future speech $x_{t+\delta}$ for the purpose of computing attacks is infeasible.

Rather than forecasting the signal, we will learn to forecast the attack vector, which encloses all possible variations of the next utterances conditioned on the current input. This attack will learn to "hedge the bet" by finding a single, minimal pattern that robustly obstructs all upcoming possibilities. Under the perturbation bound $\epsilon$, we model predictive attacks as:

$$\alpha_{t+\delta+r} = g_\theta\left(x_t\right) \quad \text{s.t.} \quad \left\|g_\theta\left(x_t\right)\right\|_\infty < \epsilon, \tag{1}$$

where $g_\theta$ is a predictive model conditioned on the present input speech and paramaterized by $\theta$. To be consistent with our notation, which represents $x_t$ as the signal until time $t$, we include an additional offset $r$ to represent the temporal duration of the attack. To satisfy the constraint on the perturbation bound, we use the tanh activation function to squash the range to the interval $[-1, 1]$ before multiplying the result by a scalar $\epsilon$. This $\epsilon$ is equal to the product of a predetermined multiplier $m$ and the maximum of the absolute value of the input speech waveform.

With predictive attacks, the algorithm for generating obstructions in real-time becomes straightforward. After the microphone observes $x_t$, the speakers need to play $\alpha_{t+\delta+r}$ exactly $\delta$ seconds later. Since sound is additive modulo reverberation, this will cause a third-party microphone to receive the corrupted signal $x_{t+\delta+r} + g_\theta(x_t)$. We found modeling the room acoustics was unnecessary because significant reverberation already breaks state-of-the-art ASR models.

We will use neural networks to instantiate the predictive model $g$. To obtain real-time performance, our feed forward calculation needs to be less than the delay $\delta$ into the future. On commodity hardware today, this calculation is on the order of 50 milliseconds.

### 3.3 LEARNING

We will learn the parameters $\theta$ of our predictive model from a large-scale labelled speech dataset, for a specific automatic speech recognition model $f_\psi$. We formulate this as the maximization problem:

$$\max_\theta \ \mathbb{E}_{(x_t, y_t)} \left[ \mathcal{L} \left( \bar{y}_t, y_t \right) \right] \quad \text{s.t.} \quad \bar{y}_t = f_\psi \left( x_t + g_\theta \left( x_{t-r-\delta} \right) \right) \quad \text{and} \quad \| g_\theta \left( x_t \right) \|_\infty < \epsilon \quad (2)$$

where $\bar{y}_t$ is the result of the ASR model after our attack to $x_t$. The objective will drive the model to find attacks that, in the future of the signal, will maximize the expected loss of the ASR model. We optimize $\theta$ using stochastic gradient descent while keeping $\psi$ fixed. Once training is performed offline, inference is efficient, requiring just a single feed-forward computation.

### 3.4 IMPLEMENTATION DETAILS

The input to our network $g_\theta$ is the Short-Term Fourier Transform (STFT) of the last 2 seconds of the speech signal. The network outputs a waveform of 0.5 seconds, sampled at 16kHz. To calculate the STFT, we use a hamming window length of 320 samples, hop length of 160 samples, and FFT size of 320, resulting in an input dimension of $2 \times 161 \times 204$. We use a 13 layer convolutional network. The appendix has full network details.

We also experimented with a network that ouputs an STFT instead of waveform. However, regressing an STFT has no guarantee that there will be a corresponding waveform to it, which means we can not actually play it in practice. In order to prevent this from happening, there has been work that adds an additional term to minimize the loss between the predicted STFT and the nearest valid STFT (Marafioti et al., 2019). However, we found that predicting the waveform directly was both simpler and more effective.

Speech datasets generally do not have time stamps to their transcriptions. In order to train our model, we need to compute the loss between the predicted speech and the ground-truth speech, meaning that in training, we need to attack the entire speech signal, not just a small segment. We therefore need to schedule our forward and backward passes such that we have computed the attack for the entire segment before we calculate the gradients. We optimized our predictive network $g_\theta$ for 4 epochs with batch size 32 across 8 NVIDIA RTX 2080 Ti GPUs on the 100-hour LibriSpeech dataset. This computation took approximately 2 days. The learning rate started at $1.5 \cdot 10^{-4}$ and decreased using an exponential learning rate scheduler, with a learning anneal gamma value of 0.99. Our code was written in PyTorch (Paszke et al., 2019) and PyTorch-Lightning (Falcon et al., 2019).

## 4 EXPERIMENTS

The objective of our experiments is to analyze predictive attacks under the constraints of real time speech streams. We first introduce the experimental setup, baselines, and defense models. We then present our experimental results with both quantitative and qualitative evidence.

### 4.1 SPEECH RECOGNITION MODELS AND DATASETS

**DeepSpeech:** We first consider the DeepSpeech automatic speech recognition system (Hannun et al., 2014), which is a commonly used pretrained model. We build off of an open-sourced implementation and use pretrained model checkpoints.[1] Since we train and test on the model, this evaluates the white box behavior of our approach.

**Wav2Vec2:** Additionally, we evaluate our approach in a black box setting by generating attacks for speech signals and then passing them through the Wav2Vec2 automatic speech recognition model (Baevski et al., 2020), without retraining our predictive model with Wav2Vec2. We use the implementation from HuggingFace (Wolf et al., 2019).

**LibriSpeech Dataset:** We train on the LibriSpeech clean 100 hour dataset, validate on the LibriSpeech clean development set, and test on the LibriSpeech test set. For our approach, we restrict the amplitude of our predicted attack to be 0.008 times the maximum of the absolute value of the amplitude of the speech signal. We call this the relative amplitude throughout the paper. Intuitively,

---

[1]https://github.com/SeanNaren/deepspeech.pytorch

Table 1: Under real-time constraints, we quantitatively evaluate our attack method and baselines with and without defense mechanisms. We show results for both white box (DeepSpeech) and black box (Wav2Vec2) settings.

| Approach | Run Time(s) | Noise Mult. m | DeepSpeech WER | CER | +LangModel WER | CER | +Denoiser WER | CER | +AdvTrain WER | CER | Wav2Vec2* WER | CER |
|---|---|---|---|---|---|---|---|---|---|---|---|---|
| No Attack | 0 | 0 | 11.3 | 3.6 | 9.6 | 3.2 | 12.1 | 4.0 | 18.7 | 6.7 | 3.8 | 0.8 |
| Uni. Noise | 0.0006 | 0.008 | 12.8 | 3.9 | 11.3 | 2.5 | 12.2 | 4.0 | 19.4 | 4.4 | 3.9 | 0.8 |
| Uni. Noise | 0.0006 | 0.05 | 28.4 | 12.0 | 17.9 | 4.1 | 12.2 | 4.1 | 19.3 | 4.3 | 22.0 | 4.9 |
| Uni. Noise | 0.0006 | 0.1 | 47.1 | 23.3 | 30.7 | 6.9 | 12.2 | 4.1 | 19.3 | 4.4 | 70.3 | 15.9 |
| Online PGD | 3.13† | 0.008 | 20.5 | 7.8 | 17.2 | 6.8 | 27.7 | 11.8 | 22.5 | 8.4 | 44.4 | 10.1 |
| Ours | 0.014 | 0.008 | **80.2** | **51.4** | **87.6** | **49.0** | **47.0** | **24.5** | **52.5** | **29.0** | **28.0** | **6.4** |
| Offline PGD | 3.13† | 0.008 | 100.9 | 68.4 | 100.5 | 67.9 | 28.0 | 12.0 | 82.8 | 52.5 | 94.5 | 21.6 |

The † indicates a lower bound because running PGD on the denoiser takes twice the amount of time.
The * indicates the attack works on black-box models, except for the PGD baselines.

our attack sounds similar to the sound of a quiet air-conditioner in the background. We additionally evaluate on several baselines, including various levels of white noise as well as projected gradient descent. For some of baselines, we experimented with making the amplitude louder, but never below the amplitude of our predicted attack. In order to measure the time taken fairly, we measured the time necessary to create the attack vector for an input of two seconds averaged over 200 runs.

## 4.2 ATTACK METHODS AND METRICS

To evaluate our approach, we compare against several methods to obstruct the speech signal.

**Uniform Noise:** One straight-forward way to obstruct speech is to play white noise. We use the same amplitude that our attack uses. We also experimented with amplitudes that are an order of magnitude louder than our attack.

**Offline Projected Gradient Descent (PGD):** Projected gradient descent is the standard method for attacking speech samples (Madry et al., 2019). It calculates the gradient of the attack using back-propagation, adds this gradient multiplied by a step size to the attack vector, and projects this sum back into the valid bound by clipping if the gradient exceeds the designated range. For the DeepSpeech model, we run 10 iterations of projected gradient descent on the input speech signal with step size equal to 20% of the bound. For the denoiser, we ran gradient descent for 30 iterations. Since projected gradient descent requires access to the entire signal and cannot be conducted online, we use this baseline to understand what the best attack could be if we had access to the future.

**Online Projected Gradient Descent (PGD):** Offline projected gradient descent does not work in real-time, since PGD requires the entire input signal in order to optimize the attack vector, and by the time the input signal is recorded, it has already passed. To make an online version of PGD, we calculate the PGD from the window of the input stream in the same manner as described for the offline method, but apply it to the future time. We note that this is unfair to our own approach, because PGD is at least two orders of magnitude slower than our approach.

**Our Approach:** We finally evaluate our approach, which requires just a single forward pass per half second of input speech. Our attack takes 0.014 seconds for a single forward pass, meaning that we need to be forecasting at least that amount into the future. We experimented with several options for how far into the future we forecast, using larger delays to allow time for recording speech and play back of attack (0.5s, 0.75s, 1.0s). We found that 0.5s performed best empirically.

The most common way to evaluate speech recognition models is through word and character error rates. We evaluate our attacks at their capability to *increase* errors. **Word Error Rate (WER)** measures the proportion of words that were incorrectly predicted, defined as $(S_w + D_w + I_w)/N_w$, where $S_w$ represents the number of word substitutions, $D_w$ the number of word deletions, $I$ the number of word insertions, and $N_w$ is the number of words. **Character Error Rate (CER)** measures the proportion of characters that were incorrectly predicted, which is important to analyze because our attack might just disrupt a single letter but the word is still intact. It is defined as $(S_c + D_c + I_c)/C_c$, where $S_c$ represents the number of character substitutions, $D_c$ the number of character deletions, $I$ the number of character insertions, and $N_c$ is the number of characters.

### 4.3 ROBUST MODELS

We evaluate our approach with both standard automatic speech recognition models as well as their robust counterparts. There are two main methods to make models robust: input preprocessing methods and adversarial training (Zelasko et al., 2021). The former fortifies the mode by attempting to clean the data of the attack, and the latter by strengthening the robustness of the model against attacks. We implement both methods to evaluate our approach.

**Audio Denoiser:** The standard way to suppress noise is to denoise the input signal. We use a state-of-the-art audio denoiser on the attacked inputs (Xu et al., 2020). In order to make our attacks robust to this form of preprocessing, we retrain our predictive model $g_\theta$, this time passing $\hat{y}_{x_t}$ through the denoiser model $h_\phi$, before passing it the automatic speech recognition model: $\bar{y}_t = f_\psi \left( h_\phi \left( x_t + g_\theta \left( x_{t-r-\delta} \right) \right) \right)$. Once again, we keep the automatic speech recognition model $f_\psi$ and the denoiser model $h_\phi$ fixed, while updating our predictive model $g_\theta$.

**Adversarial Training:** In addition, we use adversarial training to create a robust speech recognition system. We fine-tune the automatic speech recognition model on the adversarial examples. To maintain the performance on clean examples, we also add regular inputs in our training, following standard practice (Zhang et al., 2019). We train the DeepSpeech model $f_\psi$, and every batch contains half clean inputs, and half attacked inputs with 3 steps of projected gradient descent. This is already more attack than the fast-adversarial training approaches (Wong et al., 2020), as they use 1 step, making this model very robust. We call the robust model $f'_\psi$. There is always a trade-off between robustness and clean accuracy (Tsipras et al., 2019). We stop training once the WER on the attacked inputs dropped sufficiently, from $100.9\%$ to $82.8\%$, and when the WER on the clean inputs increased from $11.3\%$ to $18.7\%$. As is standard practice with adversarial training, we retrained our predictive model $g_\theta$ with the new robust $f'_\psi$, giving $g'_\theta$.

**Language Model:** We also added a language model during decoding to help defend against the attacks. We use the same language model that comes standard with DeepSpeech. The language model can act as a prior to both constrain transcriptions to natural words and using the sentence context to help correct word errors.

### 4.4 QUANTITATIVE ANALYSIS

Table 1 shows that our predictive attack is able to significantly disrupt automatic speech recognition systems. When we evaluate with the standard model (DeepSpeech column), the predictive attack is able to produce a WER that is over four times more effective than a standard online PGD attack. The white noise is able to corrupt the signal, but it requires substantially more amplitude than our approach. Even when the white noise amplitude is an order of magnitude larger than ours, our method is still more effective. We see a similar rate with the CER, suggesting that its completely corrupting words. The performance of the offline attack shows that observing the future is able to further improve the error rate, at the cost of not being able to be real-time.

We furthermore found that our attack does not significantly impact the clarity of the speech for humans to understand it. We asked human subjects to manually transcribe both attacked inputs and non-attacked inputs, and we found people only made $4\%$ more errors when our attack is present.

**Results with Language Model:** Many ASR models use a language model during prediction, and we also evaluate our attack in that setting. For clean words, our results show that as expected the language model helps decrease the WER by $1.7\%$. Similarly, across the white noise baselines, the language model helps decrease the WER. However, our attack actually performed better when there is a language model present. By visualizing the transcripts, we found that our attack causes the language model to separate non-existent words into two words because the language model acts as a prior to snap the predictions to real English words. This increase in the word count causes the WER to increase. We see similar trends with CER.

**Results with Defense:** We next evaluate our model when the ASR system has a defense mechanism. As expected, the defense mechanisms cause the WER for the attacked inputs to go down for all attack approaches. However, our attack still outperforms the baselines by nearly double in both cases. The denoiser is particularly effective at removing the white noise, which is expected as this is what it is trained to do. Moreover, the state of the art methods for audio denoising, published just last year

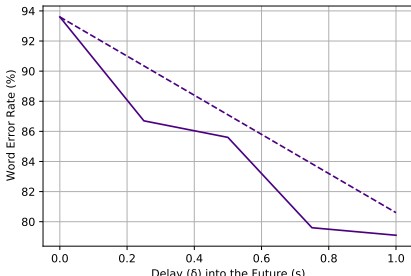

Figure 4: Delay vs. Word Error Rate

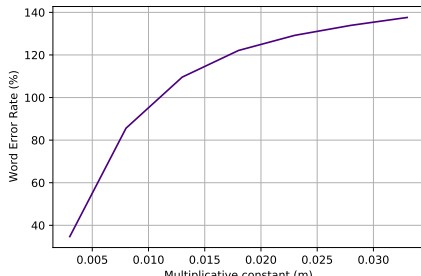

Figure 5: Multiplier vs. Word Error Rate

(Xu et al., 2020), are actually very effective at removing PGD attacks. However, our approach still manages to fool the denoiser.

We also ran inference on the adversarially trained DeepSpeech. Training the ASR model with the defense increases the WER on clean inputs by only $7.4\%$ and the CER $3.1\%$, and decreases the WER on attacked inputs by $20.7\%$ and the CER by $31.6\%$. This shows that our method is still strong as it outperforms baselines at least twice as much even without retraining our own predictive model when the automatic speech recognition model has been retrained to be more robust.

**Results with Black-box:** We also evaluated our attack when the ASR model is different from the one during training, which corresponds to a black-box attack. We apply our attack to the Wav2Vec2, and the results show that our attack is still effective. With the same attack multiplicant, we outperform the white noise baselines on WER significantly ($3.9\%$ compared to $28.0\%$). Furthermore, the only baseline that outperforms our attack is the online PGD approach, but unlike ours, this baseline is a white-box attack and consequently had access to more information.

## 4.5 CHARACTERIZING THE ATTACK

In order to analyze how our model produces attacks, we performed several quantitative experiments.

**Does the model attack specific instances?** Our attack model learns which frequencies to produce at each time in order to maximize the error rate of the ASR system. To investigate whether the attacks were adaptive to the current input, we swapped attacks from different speakers. Since input speech signals have varying lengths, the attacks will also have varying lengths. For the attack that is shorter than the speech, we repeat the attack until the entire speech covered. Conversely, if the attack is longer than the speech, then we cut it short. Our results shows that by swapping attacks for instances, the WER and CER both drop. In doing so, the WER drops from $80.2\%$ to $35.0\%$ and the CER drops from $51.4\%$ to $19.9\%$. This indicates that the model is predicting corruptions that are adaptive to the specific input.

**How robust is the attack to temporal shifts?** In practical settings, the attack may not launch at *exactly* the right time due to various delays in software and hardware systems. Therefore, we analyze how the delay $\delta$ influences the success of our attack. The larger the delay $\delta$, the further into the future our model needs to predict. We train a model to predict $\delta = 0$ into the future, and Figure 4 shows that, when it is applied for a larger delay $\delta$, the WER drops. There are two factors that could explain this drop. The first is that as the delay increases, there is a shorter amount of time for which the speech is attacked. The second factor is that as the delay increases, we are predicting further into the future, thus the future becomes more uncertain. In order to disentangle these two factors, for each different delay, we linearly scale the error rate in proportion to the decrease of time for which the attack is active (shown in the dashed line). The decreasing plot shows that the attack is sensitive to the timing even when we factor in the reduced time to deploy it. This suggests the model is learning to predict key features about the upcoming speech. However, the performance drop is not severe, showing our approach is relatively robust during inaccurate timing.

**How does power impact attack performance?** In offline settings, normal adversarial attacks aim to reduce the volume as much as possible. However, this is more challenging in over-the-air settings because there may be background ambient noise and the rogue recording device may be far away. To investigate which level of amplitude a person should select, Figure 5 shows there is a linear relation between the power and the error rate until a critical point at about $0.02$.

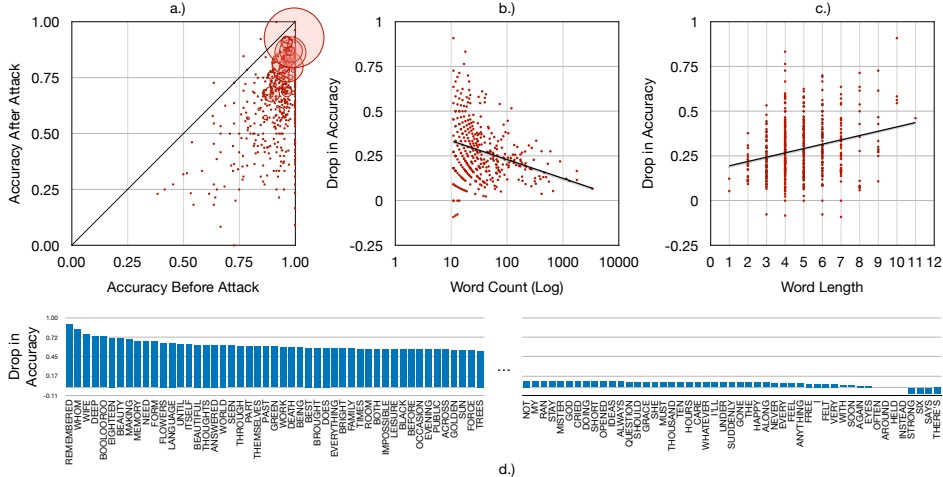

Figure 6: We analyze which words are the easiest and hardest to attack. **a**) For each word, we plot clean accuracy against the attacked accuracy. The size of the circle is proportional to the count of the word. **b**) We plot the log word count versus the drop in accuracy, with a black logarithmic trend line. **c**) We plot the drop in accuracy (due to attack) versus the word length. **d**) We plot a histogram of the easiest 50 words to attack and the hardest 50 words to attack.

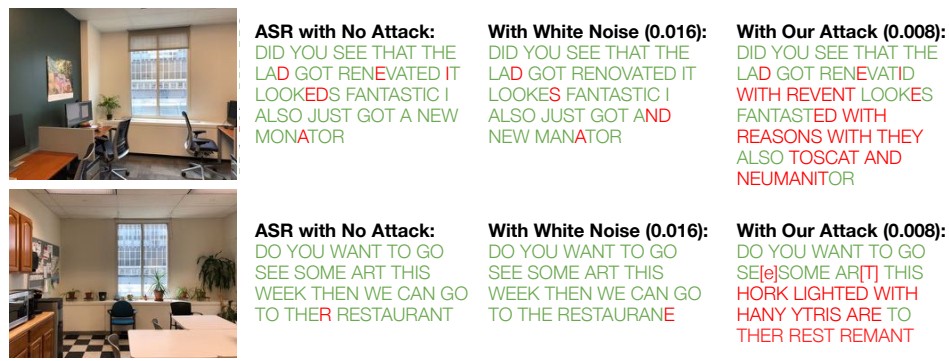

**ASR with No Attack:**
DID YOU SEE THAT THE LAD GOT RENEVATED IT LOOKEDS FANTASTIC I ALSO JUST GOT A NEW MONATOR

**With White Noise (0.016):**
DID YOU SEE THAT THE LAD GOT RENOVATED IT LOOKES FANTASTIC I ALSO JUST GOT AND NEW MANATOR

**With Our Attack (0.008):**
DID YOU SEE THAT THE LAD GOT RENEVATID WITH REVENT LOOKES FANTASTED WITH REASONS WITH THEY ALSO TOSCAT AND NEUMANITOR

**ASR with No Attack:**
DO YOU WANT TO GO SEE SOME ART THIS WEEK THEN WE CAN GO TO THER RESTAURANT

**With White Noise (0.016):**
DO YOU WANT TO GO SEE SOME ART THIS WEEK THEN WE CAN GO TO THE RESTAURANE

**With Our Attack (0.008):**
DO YOU WANT TO GO SE[e]SOME AR[T] THIS HORK LIGHTED WITH HANY YTRIS ARE TO THER REST REMANT

Figure 7: We show how our attack works in realistic rooms with diverse acoustic environments. For our attack, we use a relative amplitude of 0.008, and 0.016 for white noise.

**What makes a word easy or hard to attack?** For each word, Figure 6a plots the recognition accuracy both before and after the attack. Since most of the points are below the diagonal line, this shows our attack is effective for most words. However, some words drop in accuracy more than others. Figure 6b compares this drop in accuracy to how often the word is spoken in the dataset. The results show that the most common words (e.g. "the", "our", "they") are the most difficult to disrupt. However, by definition, the common words carry low information content, thus making them less crucial to attack. Figure 6c compares the drop in accuracy versus the word length in characters, showing a positive correlation. This result suggests that longer words are generally easier for our model to attack, possibly because they have more temporal structure to predict.

**Which words are guarded by our attack?** Figure 6d displays a histogram of the drop in accuracy for the top and bottom 50 words. The word "remembered" experiences the most significant change in accuracy, nearly becoming unrecognizable. However, some shorter words, such as "held" or "often", aren't impacted by our attack. In only 4 of 1631 cases does our attack increase the WER.

## 4.6 REAL ROOM EFFECTIVENESS

Although our attack is optimized without factoring in the room impulse response function, we find it generalizes well to real-world settings. We record a person speaking in two different areas. We played the attack through speakers in the same room to include the reverberation and ambient noise in addition to our attack. A third-party device records the sum of the attack, the speech, and the ambient noise. Figure 7 shows a few examples of our attack in acoustic environments.

## ACKNOWLEDGEMENTS

We thank Oscar Chang, Dídac Suris Coll-Vinent, Basile Van Hoorick, Purva Tendulkar and Jianbo Shi for their helpful feedback. This research is based on work supported by the NSF CRII Award #1850069 and the NSF CAREER Award #2046910. MC is supported by a CAIT Amazon PhD fellowship. The views and conclusions contained herein are those of the authors and should not be interpreted as necessarily representing the official policies, either expressed or implied, of the sponsors.

## ETHICAL CONSIDERATIONS

Our research is founded on ethical considerations. We are excited about the potential for automatic speech recognition to push the frontier of technology, such as in human-computer interaction, telecommunications, accessibility, and education. We are also keenly aware that there can be negative consequences from the deployment of machine learning models in practice. Notably, automatic speech recognition systems have raised concerns with respect to privacy. Our method is designed to protect user privacy, and return the control of user speech data back to users.

One potential limitation of our approach is that it is trained on Western speech data, and may not generalize to other cultures that are linguistically and phonetically different. The method has also not been validated on different languages or people with speech impediments. As such, the dataset and results are not representative of the population. Deeper understanding of this issue requires future studies in tandem with linguistic and socio-cultural insights.

Another potential adverse consequence is that our model is not $100\%$ accurate, and people may rely on it when it might not be. For example, the model has not been validated on the full variety of automatic speech recognition systems. We acknowledge this limitation.

The authors attest that they have reviewed the ICLR Code of Ethics and we acknowledge that this code applies to our submission. We are committed in our work to abide by the ethical principles from the ICLR Code of Ethics.

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

# A    APPENDIX

## A.1    GLOSSARY OF VARIABLES

| | |
|---|---|
| **Basics** | |
| $x_t$ | The waveform from time $0$ to time $t$ |
| $\hat{y}_t$ | The estimated speech transcription given $x_t$, assuming no attack |
| $\bar{y}_t$ | The estimated speech transcription given $x_t$, assuming our attack |
| $y_t$ | The ground truth speech transcription for $x_t$ |
| **Timing** | |
| $\delta$ | The maximum amount of time our attack will take to compute |
| $r$ | The duration of the attack waveform |
| $t + \delta$ | Given $x_t$, the earliest time our attack can begin under real-time constraints |
| $t + \delta + r$ | Given $x_t$, the time our attack will end under real-time constraints |
| **Attacks** | |
| $\alpha_t$ | The attack that finishes playing at time $t$ |
| $\alpha_{t+\delta+r}$ | The attack that finishes playing at time $t + \delta + r$ (redundant with above) |
| $x_{t+\delta+r} + \alpha_{x+\delta+r}$ | The mixed signal received by an eavesdropper |
| $\epsilon$ | The $\ell_\infty$ norm bound on the attack |
| **Models** | |
| $f_\psi(\cdot)$ | The ASR neural network with parameter vector $\psi$ |
| $g_\theta(\cdot)$ | Our predictive attack model with parameter vector $\theta$ |
| $\mathcal{L}$ | The loss function (CTC loss) |

## A.2    NETWORK ARCHITECTURE

The architecture is comprised of 8 down-sampling convolutional blocks, followed by 4 up-sampling convolutional blocks, followed by a linear layer.

The down-sampling convolutional block is comprised of a reflection padding, followed a 2d convolution layer, followed by a 2d batch norm, followed by a prelu activation function. The first downsampling block has 64 channels, the second through the seventh have 128, and finally the last one has 256 channels. The last conv block also has a leaky relu instead of prelu. Here the signal is reshaped into a one-hot vector.

The upsampling blocks are comprised of 1-dimensional ConvTranspose 1d, and a leaky relu activation function. The first has 64 channels, the second 32, the third 16, the fourth, 1. Finally, the linear layer is followed by a tanh activation function.

## A.3    RETRAINED DELAY

While Figure 6 plots the resulting WER as we shift the delay $\delta$ on a fixed trained network, below we retrain a new network per $\delta$. As expected, the larger the $\delta$, the lower the WER.

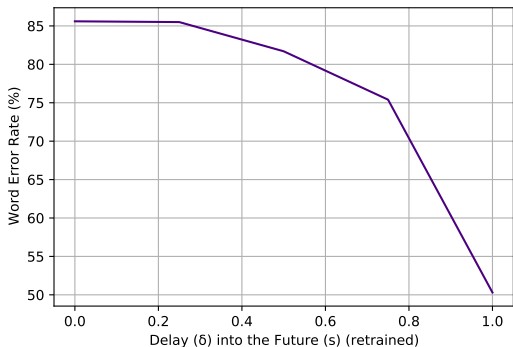

Figure 8: Retrained Delay

