# OpenReview forum: "Real-Time Neural Voice Camouflage"
_ICLR.cc/2022/Conference — ICLR 2022 Oral_

### Official Review · Reviewer_z8ss · 2021-10-25

**Correctness:** 3
**Technical Novelty And Significance:** 3
**Empirical Novelty And Significance:** 2
**Recommendation:** 8
**Confidence:** 4

**Main Review:**

I think that the idea of training a model to learn predictive attacks is a contributive and effective way for an NVC model to be used in a real-world scenario. Also, the various experiments in this paper seem that the authors have considered a lot about the real-world scenario and can give many insights to the future works.

However, there are several concerns about this paper.
1. Personally, this paper was difficult to understand especially due to the way of indexing. For example, $\alpha_{t+\delta+r}$ means a noise to be added to the speech up until $t+\delta+r$?
2. I think the $\delta$ is rather a room for computation time than an exact computation time. Is it right?
3. I think it would be better if there is a comparison with a previously proposed real-time NVC method even if it works only in a certain frequency region. This is because it seems more plausible online NVC model compared to the online PGD.
4. The paragraph, "How robust is the attack to temporal shifts?" is a little difficult to understand. When I read it, I think it is not an ablation study about varying $\delta$, but I think the paragraph is saying about it (e.g. the sentence, the larger the delay $\delta$, the further into the future our model needs to predict."). Plus, I think there should be an ablation study about varying $\delta$ for training the NVC model.
5. When it comes to the real-world scenario, I think it is also a very important condition where we do not know about the ASR model. Therefore, I think it would be better to conduct experiments showing the performance drop when using an ASR model which is different from the ASR model used in training.

**Personal Opinion**
* When I read a paragraph "Real-time Machine Learning" in Section 2 at first, I felt it could not have been written because this paper is not about speeding up the inference speed. Therefore, I think it would be better if it explains how the delay enables this method to be operated in real-time.
* I think there should be a reference about the numbers appearing when it explains the relationship between the high sampling rate and instantaneous computation.
* There should be a reference about DeepSpeech, and in Section 3.4 (or in the appendix section), I think the architecture about DeepSpeech and the Language Mdeol should be written.
* When I read this paper, I really wondered about the audio samples (generated perturbation only / speech + perturbation). However, I cannot see it and I cannot open the video in the supplementary material.
* How the WER can be larger than 100%? (off-line PGD)
* I'm a little confused about using 0.5s delay, considering the summation of computation and playback time? (0.014s + 0.5s)


**Summary Of The Paper:**

This paper proposes a Neural Voice Camouflage (NVC) method that has three important characteristics, which are essential for an NVC method to be used in practical scenarios: general, real-time, and robust. Since the proposed method trains a model to learn predictive attacks without any constraints about input and output, it can be applied to any vocabulary in a real-time scenario, and it is also difficult to defend the attack. On the contrary, the previous gradient-based adversarial attacks take a lot of time to compute the attack, so it is difficult to be used in a real-time scenario. Other than that, other previous methods are trained to attack only a few target words or utilize a pre-defined frequency region that can be easily filtered out.

In experiments, this paper shows that the proposed method is really effective by showing that the WER&CER of an ASR model significantly increases with the method compared to other NVC methods. Furthermore, this paper conducts various analyses on the behavioral characteristics of the method that can give many insights for future work. Moreover, various experiments, which are conducted with considerations about the situation where the method is used in the real world, are also shown in this paper.

**Summary Of The Review:**

I think this paper proposes a contributive NVC model and can give many insights. However, I personally think that this paper is a little difficult to read and the experiments are a little weak to support its contribution. Therefore, I give a score of 5 for this paper.

---

> ### Author Response · Authors · 2021-11-19
> **Responses to Reviewer z8ss**
>
> Thank you very much for your comments. We hope we clarified your confusions below, and please let us know if you need any additional information.
>
> **Difficult to understand especially due to the way of indexing. For example, αt+δ+r means a noise to be added to the speech up until t+δ+r?**
>
> This definition is provided in the line after Equation 1 and in Figure 2. In addition, to make it more clear, we have added a section summarizing the notation in Appendix A1.
>
> **I think the δ is rather a room for computation time than an exact computation time. Is it right?**
>
> Yes, that’s correct. We updated our definition of delta to be an upper bound in the description of Figure 2.
>
> **Comparison to other NVC methods**
>
> The only online methods are the microphone jamming approach, which requires specialized hardware, or the reinforcement learning approach, which isn’t applicable since they only attack a dataset of 10 words. To understand how the reinforcement learning approach would work assuming it worked perfectly, we ran an oracle experiment. In the dataset, we replaced all the 10 words that their approach was able to target with random letters, ensuring that the length of the word did not change. We then computed the WER of this modified dataset with the unmodified dataset, which gives us the upper bound performance of the reinforcement learning approach. Our results were that the WER went from 11.3% on the unmodified dataset to 14.7% in this modified dataset, still significantly underperforming our approach. This shows that even if this baseline worked perfectly, our approach would still be preferable.
>
> **Ablation study about varying δ**
>
> Please let us know if we misunderstood, but we believe you were asking about whether we retrain a new model per delta in Figure 6 (which is now Figure 5). This plot is showing the result of shifting delta on a fixed model. This is important because the attack may not play at the right time due to physical variabilities in software & hardware. The result shows that our system is robust and therefore practical in real-life scenarios. In addition, we found your idea of retraining a model per delta interesting, and provided an additional plot to reflect this in Appendix A3. This plot shows that as we train a model with a larger delta, the WER drops, as expected.
>
> **Black Box**
>
> Thank you for your suggestion. We passed the attacked inputs through Wav2Vec2, where the attacks were generated by the forecasting model that was trained with DeepSpeech. Please see the paper for results in Table 1 and analysis in Section 4.2, as well as the response to reviewer j1oS. In summary, our result is still effective in black box settings.
>
> **Related Work**
>
> Thanks for the suggestion, we clarified that real-time machine learning includes other methods of making it real-time besides improving inference speed.
>
> **High sampling rate/Instantaneous computation**
>
> This is just a simple calculation that if the sampling rate is 16KHz, then the time it takes for a single sample to be recorded is the reciprocal of that, which is 0.0000625 seconds, and this is within milliseconds.
>
> **There should be a reference about DeepSpeech, and in Section 3.4 (or in the appendix section)**
>
> We apologize for omitting these details. We added a new subsection, 4.1, which describes the DeepSpeech model, the implementation we built off of, and the dataset. We also describe the Wav2Vec2 model and its implementation.
>
> **When I read this paper, I really wondered about the audio samples (generated perturbation only / speech + perturbation). However, I cannot see it and I cannot open the video in the supplementary material**
>
> We have converted our demo to mp4, and reuploaded it. Please let us know if you are still unable to access it.
>
> **How the WER can be larger than 100%? (off-line PGD)**
>
> This can happen if the predicted sentence contains more incorrect words than the ground truth sentence.
>
> **I'm a little confused about using 0.5s delay, considering the summation of computation and playback time? (0.014s + 0.5s)**
>
> .014 has to be less than 0.5s. We chose a larger delta 0.5s in order to give enough room for variance, even though our forward pass is only 0.014s.

---

> > ### Comment · Reviewer_z8ss · 2021-11-23
> > **Response to the authors**
> >
> > Thank you for your considerate responses to my review and for conducting extensive additional experiments in a short period of time. They have resolved most of the concerns.
> >
> > In addition, I also propose several minor comments that might be added to future revisions.
> > * I recommend removing the human's head in Figure 2. It confused me to think that the signal farther to the head is pronounced earlier.
> > * The explanation and the added glossary helped me understand this paper more comfortably. However, I still think it might be easier if the indexing becomes more precise. (e.g. express the generated noise in Figure 2 as $\alpha_{t-r:t}$  instead of $\alpha_{t}$)
> >
> > To sum up, I think this paper is much more improved thanks to the authors and other reviewers, so I will raise my score from 5 to 8.

---

### Official Review · Reviewer_j1oS · 2021-10-31

**Correctness:** 4
**Technical Novelty And Significance:** 3
**Empirical Novelty And Significance:** 3
**Recommendation:** 8
**Confidence:** 4

**Main Review:**

### Strengths

- The motivation for the problem, the formalization, and the experiments are clearly written and well-organized.
- The proposed method is tested in various situations that reflect real-world scenarios, and successfully deployed the method in a real room environment.
- The authors also show that the attack is specific to the speaker; which partially implies that the NN-based approach is crucial for a given problem.

### Limitations

- Though practical ASR applications use an additional LM (language model) to correct the output, such cases are not examined. By looking at some examples of attacked transcription and the ground truth labels, I'm pretty sure that the accuracy of the model will increase if aided with LM. This is the main reason I'm giving a score of 5, and I'm willing to adjust my judgment if authors succeed in examining and discussing the effect of LM.
- One of the limitations of this work is that it was only tested with a single specific ASR model; I think this should be also mentioned in the "Ethical considerations" section.

### Questions

- What DeepSpeech model are the authors referring to? Please be specific about the model information and cite the literature if necessary; the authors might want to add a section in the appendix for this.
- In Figure 6, I wonder if the authors forgot to multiply 100 on WERs. It'll be clearer if "(%)" is added to every WER/CER in plots.

**Summary Of The Paper:**

A novel technique that prevents the ASR (DeepSpeech) from correctly recognizing the speech is presented. The proposed method works in real-time and is robust against some defenses.

**Summary Of The Review:**

Though this work has established the important problem and nicely tested the proposed method, it has failed to address an important component of ASR, LM. Thus I'm initially giving a score of 5.

---

> ### Author Response · Authors · 2021-11-19
> **Responses to Reviewer j1oS**
>
> Thanks for your helpful experiments suggestion ideas! We ran them and report the results below. Please let us know if you need any additional information.
>
> **Language Model**
>
> Thank you for your suggestion. We ran the experiment you suggested for our approach as well as all baselines. We used the same Language Model that DeepSpeech uses out of the box. We added these results in a new column in Table 1, and we also included a paragraph describing our results in Section 4.2. In summary, these new results show that the attack still works and outperforms baselines.
>
> **Black Box**
>
> Thank you for your suggestion. We also ran the experiment you suggested and tested our approach in a black box setting by sending the attacked inputs through Wav2Vec2. We found it still outperforms other black box baselines. Online PGD does outperform our method in this case, however that is a white box attack.
>
> **Ethics Statement**
>
> We also added a new paragraph to the ethics statement. Please let us know if there’s something missing.
>
> **Model & Dataset Details**
>
> We apologize for omitting these details. We added a new subsection, 4.1, which describes the DeepSpeech model, the implementation we built off of, and the dataset. We also describe the Wav2Vec2 model and its implementation.
>
> **Figure 6 & WER**
>
> Yes, good catch. We have updated the paper with this suggestion.  Please note, Figure 6 became Figure 5.

---

> > ### Comment · Reviewer_j1oS · 2021-11-20
> > **Thank you for your response**
> >
> > I am raising my score to 8 since my concerns are resolved and the paper has enhanced further, thanks to the other reviewer's suggestions and the author's delicate revision.
> >
> > I'd love to see the paper presented at ICLR.

---

### Official Review · Reviewer_9gbp · 2021-11-02

**Correctness:** 3
**Technical Novelty And Significance:** 3
**Empirical Novelty And Significance:** 3
**Recommendation:** 8
**Confidence:** 4

**Main Review:**

Strengths:
- the paper is well structured and easy to follow
- clearly explains how the proposed method works
- evaluation framework is well designed and straight forward
- experiments are solid and the results support the proposed method is working
- in-depth analyses on the results:
I really enjoyed the analysis shown in Figure 7

Weaknesses:
- some arguments are not validated:
In Section 4.4, the authors provide in-depth analyses and discussions on the attack characteristics. The first was whether the proposed model attacks vocal timbres, and concludes it does and attacks are speaker-dependent by stating that the attack performance drops - i.e., both WER and CER drop - when swapping attacks for speakers. However, it may not be true unless the experiments were carefully conducted with speech samples where different speakers say the same content. It would be interesting to see the results of voice-converted samples.

- some observations are not scientifically grounded:
In Figure 3 and 4, the authors repeatedly state that the attacks resemble speech "formants" but I don't see any formant-like structures in the spectrograms. If the authors are referring to the wave-like frequency components, I'm quite confident they are not formants. If time in seconds is denoted in the x-axis and the corresponding text is aligned and overlaid, it will be easier to determine.

Some minor comments:
- supplementary video was helpful to experience how the attack sounds like, but it was pretty disturbing. And the white noise was inaudible so I couldn't make comparison. If such attacks are practically to be used, it would be good to perform a user study for perceptual evaluation of different attack methods.
- is multiplier m in Figure 6 the same as Power of Noise in Table 1? And what is the unit of WER in Figure 6? Why such big discrepancy in WER?

**Summary Of The Paper:**

This paper proposes a novel attack approach with a purpose of disrupting the automatic speech recognition system. The proposed method, called Neural Voice Camouflage, works in real time by forecasting attacks ahead of time when they are added to speech streams. The authors conducted experiments with the LibriSpeech dataset, and showed that the proposed model outperforms the conventional methods with or without defense mechanisms on the task of speech recognition (performance measured by WER/CER).

**Summary Of The Review:**

This paper presents Neural Voice Camouflage, a real-time attack method that disrupts in streaming ASR systems. The methodology is clearly explained and the main contributions are well supported by a solid evaluation framework and carefully designed experiments. Thorough analyses on the results provide useful insights for researchers working in the same domain.

---

> ### Author Response · Authors · 2021-11-19
> **Responses to Reviewer 9gbp**
>
> We appreciate your thoughtful review and we hope we addressed your concerns. Please let us know if you'd like any further information.
>
> **Vocal timbres & formants**
>
> Thank you for pointing these points out. These are good points and you are right. We have modified the paper to reflect this. Specifically, we revised the claim to now say that our approach attacks instances rather than vocal timbres. Additionally, we removed our statement that the attacks resemble formants.
>
> **User study**
>
> Thank you for your suggestion. We conducted a user study to investigate this. We found that our attack does not significantly impact the clarity of the speech for humans to understand it. We asked human subjects to manually transcribe both attacked inputs and non-attacked inputs, and we found people only made 4% more errors when our attack is present. We have included this analysis in section 4.4 of the paper.
>
> **Is multiplier m in Figure 6 the same as Power of Noise in Table 1? And what is the unit of WER in Figure 6? Why such a big discrepancy in WER?**
>
> Sorry for this confusion. We have updated Table 1 to make it clear that the values were multiplicants, not power. We updated Figure 6 (which is now Figure 5) to clarify that the units for WER are percentages. The big discrepancy is because as you make the attack louder, the signal-to-noise ratio gets worse, breaking the ASR model even more.

---

### Decision · Program_Chairs · 2022-01-20

**Decision:**

Accept (Oral)

**Comment:**

This paper proposes a novel neural voice camouflage method that learns predictive attacks without any constraints about input and output. It is general, robust, and real-time that could be used in a real-world scenario. The experiments are solid, the in-depth analyses are convincing.